# Selection of Optimal Diagnostic Positions for Early Nutrient Deficiency in Cucumber Leaves Based on Spatial Distribution of Raman Spectra

**DOI:** 10.3390/plants14081199

**Published:** 2025-04-12

**Authors:** Zhaolong Hou, Yaxuan Wang, Feng Tan, Jiaxin Gao, Feng Jiao, Chunjie Su, Xin Zheng

**Affiliations:** 1College of Engineering, Heilongjiang Bayi Agricultural University, Daqing 163319, China; houzhaolong@byau.edu.cn (Z.H.); a441380540@163.com (J.G.); 2College of Civil Engineering and Water Conservancy, Heilongjiang Bayi Agricultural University, Daqing 163319, China; wangyaxuan1980@byau.edu.cn; 3College of Information and Electrical Engineering, Heilongjiang Bayi Agricultural University, Daqing 163319, China; bayitf@byau.edu.cn; 4College of Agriculture, Heilongjiang Bayi Agricultural University, Daqing 163319, China; jiaofeng1980@163.com; 5College of Horticulture and Landscape Architecture, Heilongjiang Bayi Agricultural University, Daqing 163319, China; scj_1992@163.com

**Keywords:** nutrient deficiency, cucumbers, Raman spectroscopy, precision agriculture, leaf position, nitrogen, phosphorus, potassium

## Abstract

Accurate diagnosis of crop nutritional status is critical for optimizing yield and quality in modern agriculture. This study enhances the accuracy of Raman spectroscopy-based nutrient diagnosis, improving its application in precision agriculture. We propose a method to identify optimal diagnostic positions on cucumber leaves for early detection of nitrogen (N), phosphorus (P), and potassium (K) deficiencies, thereby providing a robust scientific basis for high-throughput phenotyping using Raman spectroscopy (RS). Using a dot-matrix approach, we collected RS data across different leaf positions and explored the selection of diagnostic positions through spectral cosine similarity analysis. These results provide critical insights for developing rapid, non-destructive methods for nutrient stress monitoring in crops. Results show that spectral similarity across positions exhibits higher instability during the early developmental stages of leaves or under short-term (24 h) nutrient stress, with significant differences in the stability of spectral data among treatment groups. However, visual analysis of the spatial distribution of positions with lower similarity values reveals consistent spectral similarity distribution patterns across different treatment groups, with the lower similarity values predominantly observed at the leaf margins, near the main veins, and at the leaf base. Excluding low-similarity data significantly improved model performance for early (24 h) nutrient deficiency diagnosis, resulting in higher precision, recall, and F1 scores. Based on these results, the efficacy of the proposed method for selecting diagnostic positions has been validated. It is recommended to avoid collecting RS data from areas near the leaf margins, main veins, and the leaf base when diagnosing early nutrient deficiencies in plants to enhance diagnostic accuracy.

## 1. Introduction

As the tension between human activities and the environment intensifies, the transition from traditional extensive agriculture to precision agriculture has emerged as a pivotal direction in modern agricultural development. Precision agriculture requires rapid, reliable, and non-destructive methods to capture crop information [1]. Nutrient diagnostics play a pivotal role in this transition. Nitrogen (N), phosphorus (P), and potassium (K) are essential nutrients for plant growth, each playing critical roles in metabolism and development. They are central to crop nutrient diagnostics and fertilizer management strategies [2]. N is a fundamental component of plant proteins and chlorophyll, and its deficiency leads to leaf yellowing and growth retardation [3,4]. P plays a critical role in energy transfer and photosynthesis, with early-stage P deficiency causing dark green leaf discoloration and impairing root development [5]. K functions as an activator of various enzymes involved in photosynthesis and significantly enhances crop stress resistance. A deficiency in K results in leaf yellowing and browning, while also reducing the plant’s ability to resist diseases [6]. Although laboratory-based chemical analyses (e.g., Kjeldahl [7], spectrophotometry [8]) offer high accuracy, they are time-consuming, destructive, and risk environmental contamination. It is evident that high-throughput, cost-effective, and non-destructive detection technologies are poised to revolutionize crop nutrient diagnosis, significantly advancing the field of precision agriculture.

Raman spectroscopy (RS) is a non-invasive, non-destructive technique that resists moisture interference, does not require sample pretreatment, and allows for in vivo detection. This method is adept at identifying subtle changes in the biochemical composition of plant tissues, offering a robust tool for the early detection of plant nutritional stress [9]. Notably, RS can swiftly analyze plant samples, detecting pathogens or identifying sources of abiotic stress within seconds [10,11,12]. Importantly, experimental evidence indicates that RS lasers do not cause thermal or photodegradation of plant materials [13]. While Raman spectroscopy is effective in detecting visible nutrient deficiencies, its capability to diagnose latent starvation, where biochemical changes precede visible symptoms, remains an important research area.

Nutritional stress in crops induces changes in specific chemical components within leaf tissues, which ultimately manifest as visible deficiency symptoms [14]. Leaf-based diagnostics is a prevalent method for assessing plant nutritional status in agricultural practice [15]. While previous studies have explored the application of RS for diagnosing nutrient deficiencies in crops such as rice, Arabidopsis thaliana, Pak Choi, and Choy Sum [16,17], they have not specifically addressed the influence of leaf position selection on diagnostic accuracy during spectral data collection. Traditionally, repeated measurements or multi-point detection have been commonly employed to obtain averaged spectra, thereby mitigating the influence of anomalous spectral data [16,18]. Prior research has demonstrated significant spatial variability in nutrient distribution within the same leaf [19]. In comparison, systematically determining the optimal diagnostic positions on individual leaves is of greater practical significance [20,21]. Moreover, several reports have attempted to enhance the accuracy of RS diagnosis through combinations of various preprocessing methods [22], feature extraction techniques [23], or algorithmic improvements [24]. Nevertheless, these approaches often overlook a critical issue: the validity of the results heavily depends on the accuracy of the raw spectral data. Therefore, identifying specific positions on the leaf that yield the most stable and representative Raman spectra is critical for improving diagnostic accuracy in early nutrient deficiency detection.

Cosine Similarity Analysis (CSA) is a metric used to evaluate the cosine of the angle between two vectors within a vector space. Due to its computational simplicity and independence from absolute intensity, this method has been widely applied in fields such as spectral analysis and anomaly detection [25,26,27]. CSA not only reflects the similarity between vectors, but also captures the variations in their individual components, making it a useful tool for identifying relevant or anomalous samples in complex datasets [28].

In this study, we focused on the selection of optimal positions for the early diagnosis of N, P, and K deficiencies in cucumber leaves. Nutrient stress was induced in cucumber plants through soilless cultivation. Spectral data from cucumber leaves were collected using a Raman spectrometer in a dot-matrix pattern, referencing sensor array methodologies, across different stress durations and the control group [29]. Concurrently, chemical analysis was performed to quantify N, P, and K levels in the leaves, enabling the monitoring of nutrient deficiency progression. By analyzing the spatial variations in spectral similarity in different positions, we aimed to identify key positions on cucumber leaves with stable spectral responses that accurately reflect nutritional status. Finally, low-similarity spectral data were removed from each leaf, and the performance of the diagnostic model was compared before and after data cleansing to validate the effectiveness of the diagnostic positions selection method.

## 2. Materials and Methods

### 2.1. Plant Growth Conditions and Experimental Design

We hypothesized that spatial–spectral variations in leaves could be systematically analyzed to determine optimal diagnostic positions for early nutrient deficiency detection. The cucumber variety (Jin You 401), developed by the Tianjin Kerun Cucumber Research Institute, was used in this study. Perlite served as the substrate for indoor soilless cultivation, ensuring controlled and consistent cultivation conditions across all samples. The nutrient solution was based on the Hoagland formula, with normal nutrient solution applied during the early growth stages [30]. Throughout cultivation, plants were exposed to fluorescent light for 16 h per day, maintaining a relative humidity of (50 ± 5)%, and a temperature of 29 °C/26 °C (day/night). The hydroponic system was regularly monitored to maintain stable pH and electrical conductivity levels. Due to the high uniformity of cucumber seedlings during early growth and their heightened sensitivity to nutrient stress, 48 uniformly grown seedlings were selected for experimentation at the two-leaf, one-core stage. The cultivation substrate was first rinsed with deionized water before the seedlings were randomly assigned to four groups: control-check (CK), nitrogen-deficient (ND), phosphorus-deficient (PD), and potassium-deficient (KD), with twelve plants per group. Each group was cultivated with its respective nutrient solution. The CK nutrient solution followed the complete Hoagland formula, containing Ca(NO_3_)_2_·4H_2_O at 945 mg/L, KNO_3_ at 607 mg/L, (NH_4_)H_2_PO_4_ at 115 mg/L, and MgSO_4_·7H_2_O at 493 mg/L, along with trace elements, including Fe-EDTA at 2.5 mg/L, H_3_BO_3_ at 2.86 mg/L, MnCl_2_·4H_2_O at 2.13 mg/L, ZnSO_4_·7H_2_O at 0.22 mg/L, CuSO_4_·5H_2_O at 0.08 mg/L, and Na_2_MoO_4_·2H_2_O at 0.02 mg/L. In the ND nutrient solution, Ca(NO_3_)_2_·4H_2_O and KNO_3_ were omitted and replaced with CaCl_2_ at 520 mg/L and KCl at 450 mg/L, respectively [16]. In the PD nutrient solution, (NH_4_)H_2_PO_4_ was removed and replaced with NH_4_Cl at 53 mg/L to maintain N supply, while in the KD nutrient solution, NaNO_3_ at 510 mg/L was used as a substitute for KNO_3_. Sampling time points were selected based on preliminary experiments and previous studies on nutrient deficiency responses in crops [17,31]. Leaf samples were collected at 24 (24 h), 72 (72 h), 120 (120 h), and 168 (168 h) hours after stress induction. At each time point, the first-node leaves from three cucumber plants per treatment group were excised, sealed in bags, and immediately brought back to the laboratory for RS data collection and chemical analysis of N, P, and K content. A total of 12 leaf samples were collected at each time point, resulting in 48 leaf samples across all time points. Although Raman spectroscopy is inherently a non-destructive technique, in this study, leaf excision was employed to ensure consistent measurement conditions and improve data reliability. This approach provides a foundation for future in vivo non-destructive diagnostics.

### 2.2. Measurement of Leaf NPK Content

After collecting the spectral data, cucumber leaves were immediately placed in an oven at 105 °C for 20 min to deactivate the enzymes [32]. Subsequently, the leaves were dried at 80 °C to a constant weight [33]. The dried samples were then stored in a desiccator for subsequent chemical analysis.

The samples were weighed, ground, and digested with concentrated H_2_SO_4_. The N content was determined using the Kjeldahl method, with the total N content (*w*_1_) calculated using Equation (1) as follows:(1)$$w_{1}=\frac{(V_{2}-V_{0})\times c\times 0.014}{m\times (V_{1}/V)}\times 100$$
where *c* is the concentration of the sulfuric acid standard titration solution (1/2 H_2_SO_4_) in 0.01 mol/L, *V*_2_ is the volume of the standard acid solution consumed by the sample (mL), *V*_0_ is the volume consumed by the blank (mL), *V*_1_ is the volume of liquid A tested during distillation (mL), *V* is the total volume of liquid A tested (mL), *m* is the sample mass (g), and 0.014 represents the mass of N in 1 mL of 1 mol/L sulfuric acid standard titration solution (g).

The P content was measured by molybdenum-antimony anti-absorption spectrophotometry, with the total P content (*w*_2_) calculated using Equation (2), as follows:(2)$$w_{2}=\frac{\rho \times V}{m}\times \frac{V_{2}}{V_{1}}\times 10^{-4}$$
where *ρ* is the mass concentration of P in liquid A (mg/L), *V* is the total volume of liquid A tested (mL), *V*_1_ is the dispensed volume of liquid A tested (mL), *V*_2_ is the volume of the color solution (mL), and *m* is the sample mass (g).

The K content was assessed using flame photometry, with the total K content (*w*_3_) calculated using Equation (3), as follows:(3)$$w_{3}=\frac{\left(\rho -\rho_{0}\right)\times V}{m}\times \frac{V_{2}}{V_{1}}\times 10^{-4}$$
where *ρ* is the mass concentration of K in liquid A (mg/L), *ρ*_0_ is the mass concentration of K in the reagent blank digestion solution (mg/L), *V* is the total volume of liquid A tested (mL), *V*_1_ is the dispensed volume of liquid A tested (mL), *V*_2_ is the volume of the color solution (mL), and *m* is the sample mass (g).

The NPK content results were expressed as g·kg^−1^ on a dry weight basis.

### 2.3. Raman Spectroscopy Data Collection

To minimize fluorescence interference, a miniature Raman spectrometer (ATP3000P, OPTOSKY, Xiamen, China) equipped with a 785 nm fiber laser source was utilized to acquire spectral data. The spectrometer covered a wavelength range of 200–3400 cm^−1^ with a spectral resolution of ±4 cm^−1^, conforming to standard specifications. The RS parameters were as follows: laser intensity set at 400 mW, integration time of 3.5 s, with three consecutive acquisitions per point, and the mean of these three measurements was used as the representative spectrum.

During the preparation of the spectral sampling dot-matrix template, we tested templates with different hole spacings of 0.5 cm, 0.75 cm, and 1.0 cm. Due to the susceptibility of the template material to deformation during high-temperature laser cutting, the maximum achievable hole diameters for the 0.5 cm and 0.75 cm hole spacing templates were limited to 0.3 cm and 0.35 cm, respectively. In contrast, the 1.0 cm hole spacing template allowed for a hole diameter of 0.5 cm. Larger hole diameters facilitate data acquisition, but may reduce spatial resolution. Given the relatively large leaf area of cucumber plants, the template with a 1.0 cm hole spacing and a 0.5 cm hole diameter was therefore selected, as shown in Figure 1. Spectral data were collected in a dot-matrix pattern according to the template positions, focusing on the main vein side of the leaf. Due to variability in leaf size under different nutrient stresses and time points, the number of spectra collected varied slightly among groups. A total of 1367 spectra were acquired across four groups: 420 from CK, 303 from ND, 307 from PD, and 337 from KD.

### 2.4. Spectral Similarity Calculation Method

Cosine similarity is commonly used to evaluate the similarity between two multidimensional vectors. When the two vectors have the same orientation, the angle *θ* between them is 0°, and cos *θ* equals 1. Conversely, when the vectors are orthogonal, *θ* is 90°, and cos *θ* is 0. Therefore, a cosine value approaching 1 indicates that *θ* is near 0°, signifying a higher degree of similarity between the vectors. In this study, discrete spectral intensity distributions in the wavenumber domain are treated as multidimensional vectors. The cosine similarity is the inner product normalized by the norms of the vectors and can be expressed using Equation (4), as follows:(4)$$\cos\;\theta =\frac{\vec{A}\cdot \vec{B}}{\left\|\vec{A}\right\|\left\|\vec{B}\right\|}=\frac{\sum_{i}A_{i}\cdot B_{i}}{\sqrt{\sum_{i}{A_{i}}^{2}}\sqrt{\sum_{i}{B_{i}}^{2}}}$$
where the vectors *A* = [*A*_1_, *A*_2_, *A*_3_, …] and *B* = [*B*_1_, *B*_2_, *B*_3_, …] represent spectra with intensities *A_i_* and *B_i_* at the wavelength 1/*λ_i_*.

In the experiment, the presence of experimental errors in spectral data is unavoidable; averaging the spectra helps to eliminate biases that may arise from individual samples and provides a more accurate representation of the overall nutritional status of the leaf. Kelly et al. have explored the rationale of species mean values from a mathematical perspective [34]. Therefore, we used the mean spectrum of individual leaves as a reference spectrum. The optimal sampling areas for leaf nutrient diagnosis were analyzed by comparing the similarity between preprocessed spectra from different positions on the leaf and the reference spectrum.

### 2.5. Diagnostic Model Development and Evaluation Methods

The cucumber variety (Jin You 401), developed by the Tianjin Kerun Cucumber Research Institute, was used in this study. Perlite served as the substrate for indoor soilless cultivation, ensuring controlled and consistent cultivation conditions across all samples.

Data preprocessing is a critical step in spectral analysis, which helps to reduce the potential interference of instrumental errors and environmental factors on spectral data while maximizing spectral differences [35]. In this study, the iterative improved moving average method was used to correct the baseline drift due to background fluorescence, with the window size set to 31 and the iteration count to 5. After baseline correction, the spectral data were normalized in order to adjust the eigenvalues to a uniform scale, which solved potential issues arising from discrepancies in eigenvalue scales, thereby optimizing model performance.

Partial Least Squares Discriminant Analysis (PLS-DA) is a multivariate statistical technique extensively utilized in chemometrics [36,37]. Compared with traditional discriminant analysis methods, PLS-DA is adept at managing data characterized by strong correlations and multicollinearity, making it particularly effective in scenarios where the number of samples is fewer than the number of variables. To validate the diagnostic position selection method, we developed a PLS-DA model using the aforementioned preprocessing techniques for early detection of N, P, and K deficiencies in cucumber leaves.

To evaluate the model’s performance, we utilized several metrics, including precision (macro-P), recall (macro-R), F1 score (macro-F1), the number of latent variables (LVs), and the number of misdiagnoses (MDs). F1 score was selected as a key metric due to its strengths in handling imbalanced datasets and its ability to balance precision and recall through the harmonic mean. The optimal latent variable setting for the model was determined using K-fold cross-validation. A higher number of LVs may improve model fitting, but also increases the risk of overfitting, reducing generalizability. To achieve a balance, we determined the optimal LVs as the smallest number at which the model reached stable performance, preferably below 10, as suggested in previous studies [38,39]. In nutrient deficiency diagnosis, misclassifications can lead to incorrect fertilization strategies, potentially harming crop growth and reducing yield. Therefore, among these metrics, lower LVs and MDs values, along with an F1 score closer to 1, indicate superior diagnostic performance.

The spectral data from different stress durations and treatment groups were evenly divided into the training and test sets at an interval of 1 based on the collection sequence to validate the diagnostic position selection method. All data processing was conducted using Python 3.10.4, with plotting performed in Origin 2021.

## 3. Results

### 3.1. Identification of Nutrient Deficiency in Cucumber

Visual observations of leaf color and growth status across different treatment groups under varying stress durations revealed that after 72 h of stress, ND and PD plants exhibited slight stress responses in leaf color. However, by 168 h, although the leaf color in KD plants showed no significant changes, their growth rate was noticeably slower compared to CK. These preliminary observations suggest that early deficiencies of different nutrients may need to be detected through comparison at different times.

To further validate these visual observations, chemical analyses were conducted to measure the N, P, and K content in the samples from each group, as shown in Figure 2. The N content in the CK group leaves exhibited an increase from 24 h to 72 h, followed by a decline from 72 h to 168 h. In contrast, P content remained relatively stable, while K content showed a decreasing trend after 72 h. This phenomenon was associated with the functional transition of leaves from a “sink” to a “source” during development and maturation [40,41]. At all observed time points (24 h, 72 h, 120 h, and 168 h) after introduction to stress, N, P, and K content in the leaves of the stress group were lower than those of CK. As the stress duration increased, N and K content in the leaves showed a marked decrease, while the decline in P content was relatively moderate. Notably, an anomalous increase in P content was observed at 120 h. This phenomenon was likely attributed to the inherently narrow fluctuation range of P content, which, combined with individual variations among sampled plants, resulted in certain samples exhibiting higher P content at 120 h compared to their 72 h counterparts. However, despite this variation, the P content in stressed samples remained distinguishable from that of the CK group, confirming their nutrient-deficient status. Therefore, this deviation was considered to be within an acceptable range.

A comparative analysis of the chemical measurements and visual observations suggests that, within 24 h of nutrient stress in cucumber, an internal physiological response is initiated, despite the absence of significant morphological changes or color abnormalities in the leaves. This underscores the critical importance of timely intervention during the early stages of stress.

### 3.2. Evaluation and Analysis of Spectral Similarity at the Same Position

The cosine similarity of spectra is influenced by multiple factors, including the stability of the spectrometer, variations in the detection environment, and the ability of specific positions to represent the plant’s nutrient deficiency status. Analyzing the similarity of spectra collected repeatedly from the same position is valuable for assessing spectral consistency. In this study, a total of 1367 spectra were acquired from 48 cucumber leaves, so the average number of acquisitions per leaf was about 30. Given the relatively controlled conditions of the laboratory environment, we aimed to evaluate the potential impact of spectrometer stability on spectral similarity. To this end, three random positions on the left were selected, and 30 spectra were collected from each position. These spectra were compared to the mean spectrum of their respective positions to evaluate similarity. Table 1 summarizes the minimum similarity values for 10, 20, and 30 spectral collections at each position compared to the mean spectrum.

The results showed that the mean value of the minimum similarity for 10 spectra collected from the three positions was 0.99225, for 20 was 0.99169, and for 30 was 0.99157. Although the spectrometer exhibited high reproducibility, a slight decline in similarity was observed as the number of spectral collections from the same position increased. This trend should be attributed to the cumulative effects of spectrometer stability. Notably, the minimum similarity values of spectra repeatedly collected at each position, when compared with their respective mean spectra, remained above 0.991. Therefore, when identifying outliers in spectral similarity across positions, we prioritized spectral data with similarity below 0.991, enabling a more targeted focus on spectra that may be anomalous in characterizing the plant’s nutrient status. It is important to emphasize that 0.991 is not a fixed global threshold, but rather a reference benchmark for outlier analysis, aimed at minimizing biases stemming from instrument stability and enhancing the reliability of the analysis.

### 3.3. Analysis of Spectral Similarity Outliers at Different Positions

Box plots are highly effective for identifying outliers, providing the distinct advantage of remaining unaffected by them, which allows for an accurate and stable depiction of data dispersion. Additionally, box plots facilitate data cleansing processes. In this study, we systematically analyzed the cosine similarity between individual positions on cucumber leaves and the mean spectrum using a box plot, as shown in Figure 3. The results indicated that the minimum value of spectral similarity was 0.93328, the maximum value was 0.99724, the mean value was 0.99244, and the standard deviation was 0.00434. Points below the lowest value in the box plot were labeled as similarity outliers, resulting in the detection of 63 anomalous values. Table 2 presents the distribution of these outliers, while Table 3 summarizes the data distribution for spectral similarities below the overall lower quartile.

A comprehensive analysis of Table 2 and Table 3 reveals a discernible pattern in the distribution of similarity values. Notably, the number of spectral similarity anomalies and those below the overall lower quartile were highest in the 24 h group. The number of anomalies decreased gradually as the stress duration increased. This trend suggests that spectral similarity at various positions on cucumber leaves exhibits greater instability during the early developmental stages or under shorter periods of nutrient stress. Further comparison among treatment groups indicates that the spectral similarity in CK leaves was the most unstable, followed by ND and PD, with KD showing relatively more stability.

### 3.4. Analysis of Spatial Distribution Characteristics of Spectral Similarity on Cucumber Leaves

To further investigate the distribution characteristics of spectral similarity on cucumber leaves, we focused on the similarity values at positions below the lower quartile and replaced the similarity values at other positions with 1.0. By visualizing these processed data as a heat map, we clearly depicted the distribution of lower similarity positions on the leaves, as illustrated in Figure 4. In the heat map, the closer the similarity value is to 1.0, the darker the corresponding color appears. This color gradient effectively highlights the distribution pattern of spectral similarity across different treatment groups and stress durations. Notably, the spectral data with lower similarity were primarily concentrated along the leaf margins, followed by areas near the main veins and the leaf base, in both control plants and those subjected to stresses. The observed spectral variations likely correspond to biochemical adjustments in leaf tissues during early nutrient stress adaptation. These findings validate our earlier hypothesis that spectral data from individual positions on cucumber leaves may not consistently characterize early nutrient deficiencies, providing a critical reference for optimizing subsequent diagnostic models.

### 3.5. Establishment of an Early Diagnostic Model for Nutrient Deficiency in Cucumber

Figure 5 presents the complete process of raw spectral data preprocessing. Initially, the characteristic peaks in the raw spectra are obscured by the fluorescence background. To address this issue, we applied an iterative improved moving average method for baseline correction, as depicted in Figure 5B. The effect of this correction is illustrated in Figure 5C, where the corrected spectra more accurately reflect the original characteristic peaks, ensuring that all spectral curves are positioned above the zero-coordinate axis. To further examine the relationship between the corrected spectra and the original peaks, both the raw and baseline-corrected spectra were normalized, as shown in Figure 5D,E. Although the spectra between 200 and 700 cm^−1^ were not fully baseline-corrected due to strong fluorescence interference, the spectral characteristics in this range were rendered more clearly, which does not compromise the reliability of subsequent model training. In the range of 700 to 3400 cm^−1^, the spectra displayed a series of prominent peaks, with distinct characteristic peaks observed at 747, 917, 1005, 1048, 1080, 1117, 1155, 1185, 1218, 1265, 1288, 1301, 1327, 1387, 1440, 1488, 1528, 1611, 1674, and 3191 cm^−1^ in the Raman spectra of cucumber leaves.

We integrated the aforementioned preprocessing methods with PLS-DA to construct an early diagnostic model for N, P, and K deficiencies in cucumber leaves. To assess the diagnostic performance of the model, a confusion matrix was generated, with the horizontal axis representing the predicted labels of the samples and the vertical axis representing the true labels. The diagonal values indicate the number of correctly predicted samples, providing a clear visualization of the diagnostic results across different treatment groups. As depicted in Figure 6, the model effectively classified the spectral data of cucumber leaves from the test set into CK, ND, PD, and KD categories across different stress durations, providing empirical support for further optimization of the model.

### 3.6. Selection and Validation Analysis of Early Diagnostic Positions for Nutrient Deficiencies in Cucumber

To evaluate the correlation between the spectral similarity of cucumber leaves and the selection of early diagnostic positions for nutrient deficiencies, we excluded spectral data with similarity values below the lower quartile of each leaf, as these are shown in Figure 7, where the gray color indicates that the data from these positions were culled. Table 4 compares the diagnostic model’s evaluation results across different stress durations before and after data cleansing.

Before data cleansing, model evaluation metrics for cross-validation and the test set indicated that the optimal number of LVs across different stress durations ranged from 6 to 11, while the number of MDs varied between 7 and 15. Notably, in the 24 h group, the optimal number of LVs reached 11, the highest number of MDs (15) was recorded, and the F1 scores for cross-validation and the test set were the lowest, at 89.56% and 91.20%, respectively. These findings suggest that the model faces substantial challenges in the early (24 h) stress phase, potentially exhibiting a degree of overfitting. As stress duration increased, the number of MDs gradually declined, while macro-P, macro-R, and macro-F1 scores improved. In the 168 h group, F1 scores for cross-validation and the test set reached 91.45% and 96.03%, respectively, highlighting the significant impact of stress duration on the stability of spectral signals.

After data cleansing, the optimal LVs and MDs at each time point were reduced, particularly in the 24 h group, where the number of LVs decreased by 4 (from 11 to 7) and the number of MDs dropped by 7 (from 15 to 8), indicating a notable enhancement in model reliability during the early stage (24 h). Furthermore, Test set results demonstrated that the F1 scores for the 24 h, 72 h, 120 h, and 168 h groups increased by 2.10%, 2.16%, 1.53%, and 1.81%, respectively, surpassing the pre-cleaning outcomes. Cross-validation results further corroborated the trends observed in the test set, confirming that data cleansing not only optimized overall model performance, but also enhanced its generalization capability.

In summary, data cleansing significantly improved diagnostic accuracy, with the reduction in MDs (by up to 7) and the increase in F1 scores (by up to 2.16%) demonstrating its effectiveness. These findings suggest that removing low-similarity spectral data from areas near the margins, main veins, and base of cucumber leaves can significantly enhance the accuracy and reliability of the early diagnostic model for nutrient deficiencies. Consequently, it can be inferred that the level of spectral similarity on cucumber leaves is closely related to the reliability of the diagnostic positions.

## 4. Discussion

While RS has demonstrated significant potential for monitoring plant nutritional stress [42], research into identifying optimal diagnostic positions under varying nutritional conditions remains limited. Therefore, to effectively utilize RS for the early diagnosis of nutrient deficiencies, it is essential to identify key diagnostic positions that accurately represent the nutritional status of crop leaves across different conditions. Removing low-similarity data improved model accuracy by eliminating spectral noise and inconsistent signals from unstable leaf regions. The diagnostic position selection method proposed in this study effectively identifies anomalous spectral data on cucumber leaves, circumventing the limitations associated with the previous reliance on averaged spectral data. Our study found that avoiding the collection of RS data from areas near the cucumber leaf margins, main veins, and the leaf base can effectively improve the reliability of the spectral data. Furthermore, optimizing the sampling data through the analysis of the spatial distribution of spectral similarity obviously enhanced the PLS-DA model’s ability to diagnose early-stage N, P, and K deficiencies in cucumber, resulting in more stable and reliable diagnostic performance.

In contrast to Hu et al. [20], who primarily investigated variations in SPAD values across different positions on cucumber leaves, our study demonstrates significant variations in spectral similarity, particularly near the leaf margins, main veins, and the leaf base, where greater fluctuations and instability were observed. This finding aligns with Hu’s results in terms of positional variability; however, their study focused solely on identifying optimal sampling positions for N diagnosis, without addressing the influence of other key nutrients. Meanwhile, prior research has used hyperspectral imaging to diagnose N deficiency in cucumber plants by mapping chlorophyll distribution [43]. Nonetheless, this approach primarily targets later stages of chlorophyll degradation, which can be influenced by various biotic and abiotic stresses, thereby limiting the specificity and accuracy of hyperspectral imaging [16]. In comparison, RS, which directly samples stable spectral areas on the leaf, offers a distinct advantage for early diagnosis of nutrient deficiencies.

The preprocessed spectral results demonstrated that the spectral features within the 700–1800 cm^−1^ range in cucumber leaves closely resemble those observed in rice leaves, as reported by Sanchez et al. [17]. This similarity highlights the potential of RS for diagnosing early nutrient deficiencies across different plant species. The identified spectral peaks correspond to pectin [44], cellulose [45], carotenoids [46], phenylpropanoids [47,48], protein [49], and aliphatic vibrations [50]. This indicates that variations in spectral similarity at different leaf positions are closely related to the changes in their contents, which is crucial for the understanding of the diagnostic mechanism of plant nutrient deficiencies.

While we anticipate that the method proposed in this study will enhance the accuracy of RS for detecting nutrient status in various crops, further research is required to validate its efficacy. Notably, the dataset utilized here reflects the impact of a single variable on spectral similarity across different positions on leaves under hydroponic conditions. In practical agricultural settings, environmental factors are far more complex and dynamic, potentially introducing additional uncertainties into the selection of diagnostic positions. Moreover, variability in leaf size and environmental conditions may limit the generalizability of these spatial patterns. Future work should consider multi-crop trials and real-field conditions to validate the robustness of these diagnostic positions. Additionally, validating the model across diverse environmental conditions will be essential to enhance its practical use in real-world agricultural settings, ensuring broader applicability in precision agriculture.

## 5. Conclusions

This study introduces a novel analytical method for selecting early diagnostic positions of nutrient deficiencies in plant leaves by analyzing the spatial distribution characteristics of spectral similarity. We examined the impact of four nutrient statuses (ND, PD, KD, and CK) and four stress durations (24 h, 72 h, 120 h, and 168 h) on the selection of diagnostic positions. We systematically explored reliable areas for spectral data collection from cucumber leaves at the early stage of nutrient deficiency, considering aspects such as outlier identification, low-similarity patterns, and model validation. Our findings demonstrate that, under the tested nutrient stresses and stress durations, excluding RS data from areas near leaf margins, main veins, and the leaf base yields more representative and reliable spectral data. Integrating this diagnostic approach into automated sensor networks could facilitate real-time, large-scale monitoring of plant nutritional status. Additionally, this method effectively identifies anomalous spectral data on cucumber leaves, leading to a marked improvement in diagnostic model performance across all stress durations. Notably, during the early stress stage at 24 h, the model achieves higher diagnostic accuracy. This method not only enhances early nutrient deficiency detection in cucumbers but also lays the groundwork for non-destructive diagnostics in other crops, contributing to the advancement of precision agriculture.

## Figures and Tables

**Figure 1 plants-14-01199-f001:**
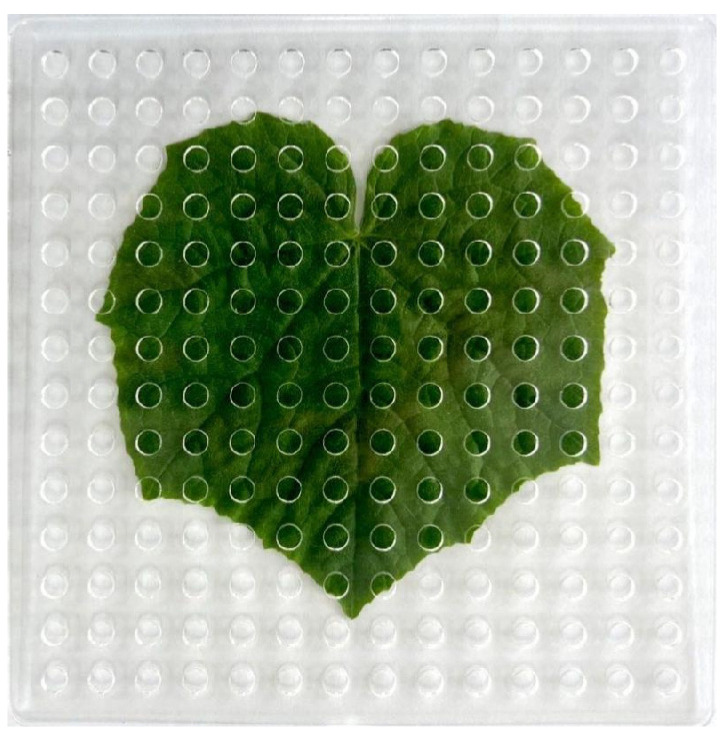
Template for spectral acquisition using a dot-matrix method on leaf surfaces.

**Figure 2 plants-14-01199-f002:**
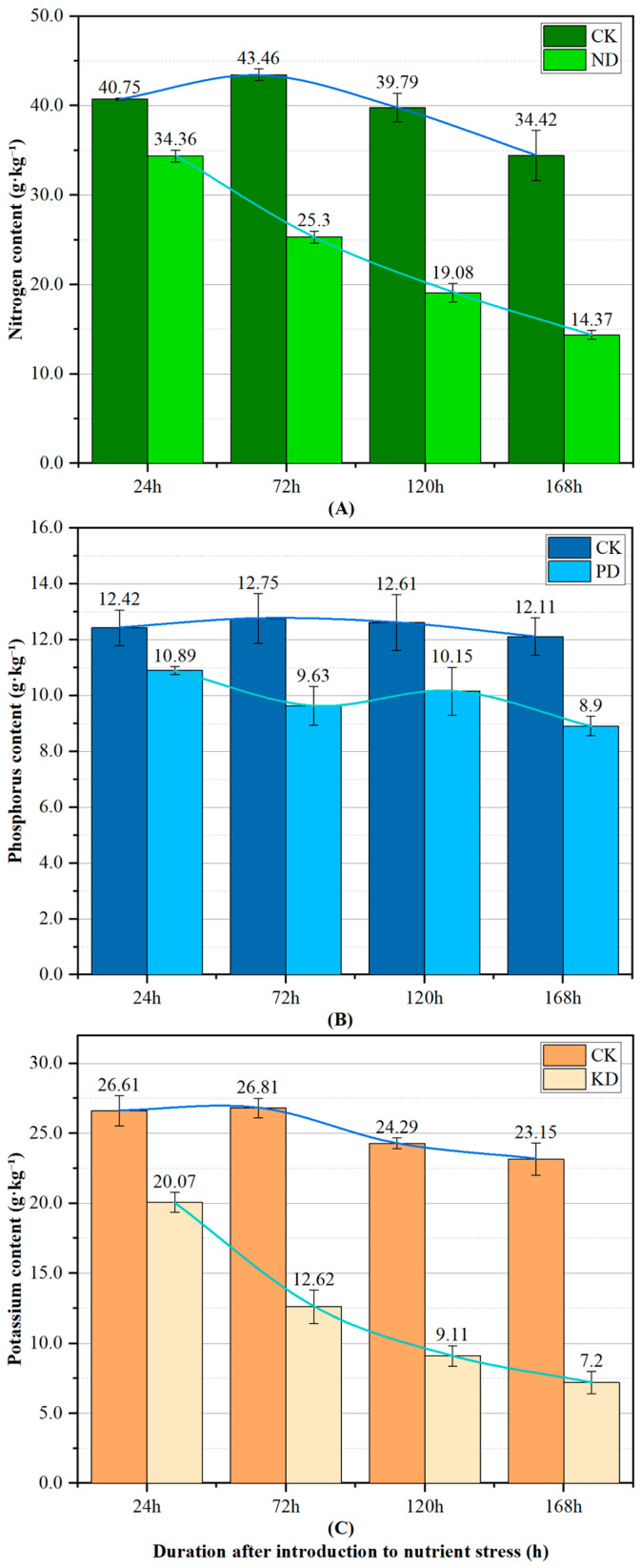
Trends in N, P, and K content in cucumber leaves over time after stress induction. Each bar represents the mean ± S.E.M. (*n* = 3). (**A**) N content in ND vs. CK. (**B**) P content in PD vs. CK. (**C**) K content in KD vs. CK.

**Figure 3 plants-14-01199-f003:**
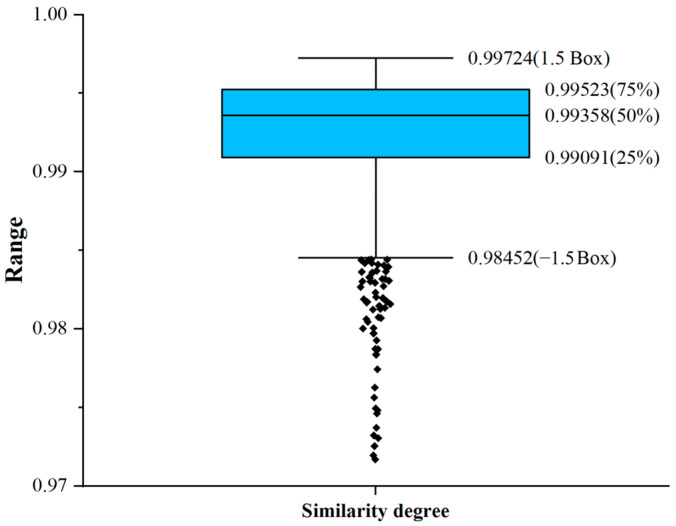
Box plot of spectral similarities across cucumber leaf positions. The boxes represent the interquartile range, and the lines inside the boxes represent the medians. The whiskers denote the lowest and highest values within 1.5 times the interquartile range, while points below the lower whisker represent outliers in similarity values.

**Figure 4 plants-14-01199-f004:**
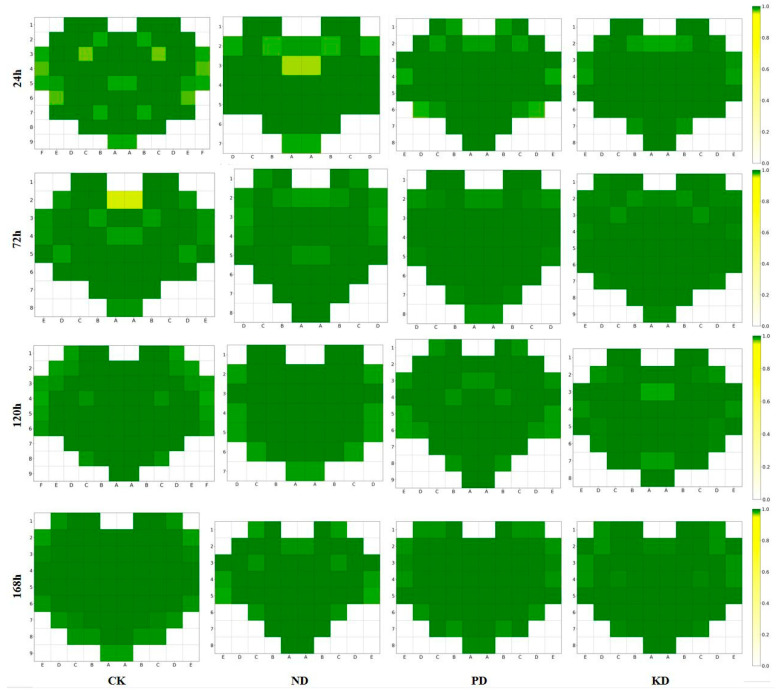
Spatial distribution of low-similarity spectral positions across cucumber leaves. CK (control-check), ND (nitrogen-deficient), PD (phosphorus-deficient), and KD (potassium-deficient).

**Figure 5 plants-14-01199-f005:**
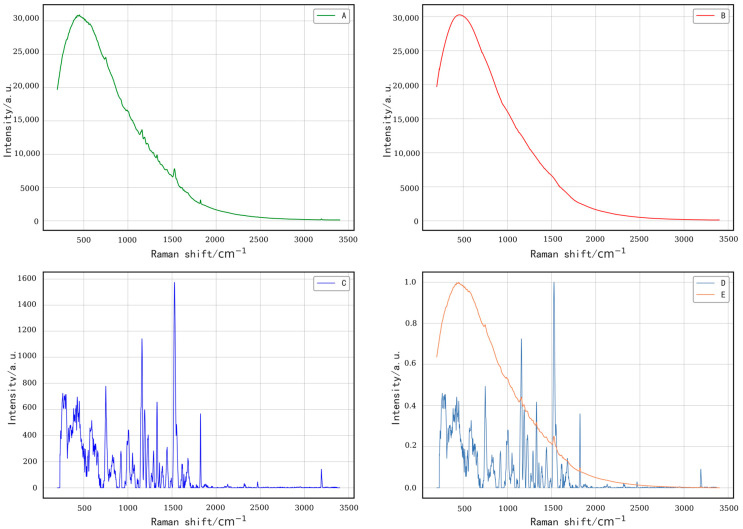
Workflow of spectral data preprocessing. (A) Raw spectra; (B) Fitted baseline; (C) Baseline-corrected spectra; (D) Normalized spectra after baseline correction; (E) Normalized raw spectra.

**Figure 6 plants-14-01199-f006:**
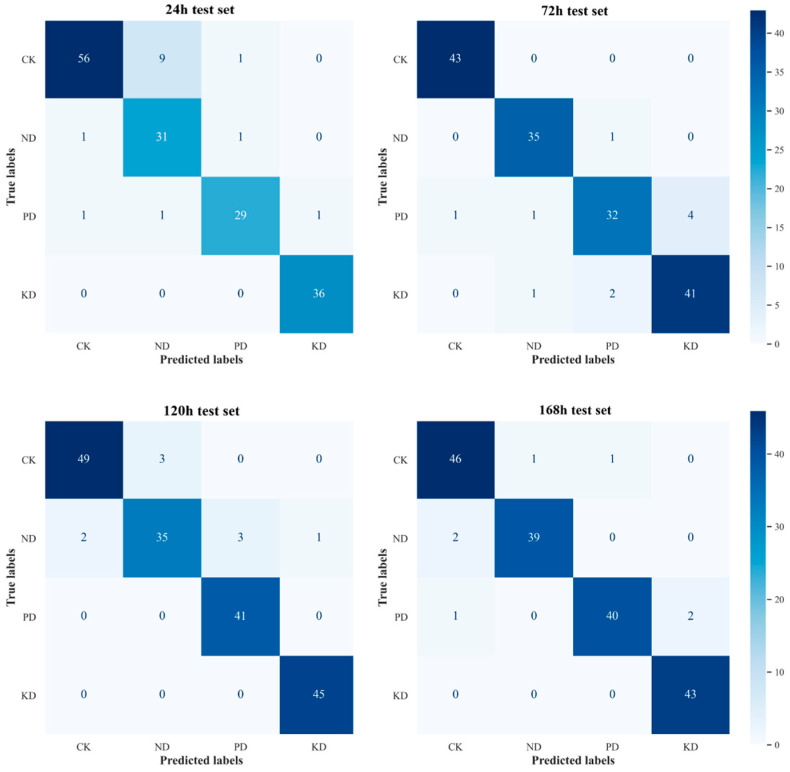
Confusion matrices for test sets under different stress durations.

**Figure 7 plants-14-01199-f007:**
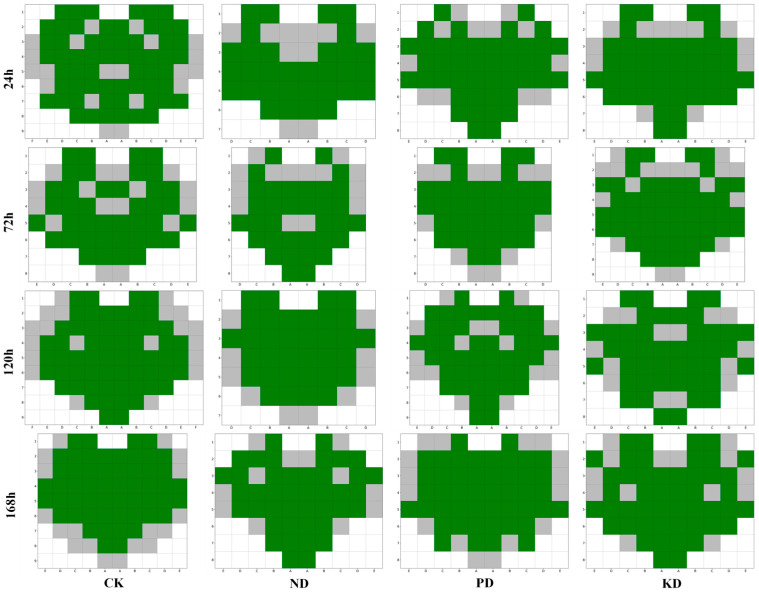
Distribution of excluded low-similarity spectral positions. Green indicates retained spectral data points, while gray indicates excluded positions.

**Table 1 plants-14-01199-t001:** Statistical summary of minimum similarity values for multiple spectral collections at the same position.

Positions	10 Times	20 Times	30 Times
P1	0.99211	0.99139	0.99135
P2	0.99191	0.99153	0.99134
P3	0.99273	0.99215	0.99203
Mean value	0.99225	0.99169	0.99157

**Table 2 plants-14-01199-t002:** Statistical distribution of anomalous spectral similarity data in cucumber leaves.

Duration of Stress	Treatment Groups				Total
	CK	ND	PD	KD	
24 h	37	9	4	1	51
72 h	2	3	0	0	5
120 h	1	2	1	1	5
168 h	1	0	0	1	2
Total	41	14	5	3	63

**Table 3 plants-14-01199-t003:** Statistical distribution of spectral similarity values below the overall lower quartile in cucumber leaves.

Duration of Stress	Treatment Groups				Total
	CK	ND	PD	KD	
24 h	107	35	23	6	171
72 h	22	28	19	7	76
120 h	21	23	11	6	61
168 h	15	11	4	4	34
Total	165	97	57	23	342

The lower quartile is 0.99091.

**Table 4 plants-14-01199-t004:** Comparison of diagnostic model evaluation metrics across different stress durations before and after data cleansing.

Group Names	Sample Size	LVs	MDs	Cross-Validation			Test Set Evaluation		
				Macro-P (%)	Macro-R (%)	Macro-F1 (%)	Macro-P (%)	Macro-R (%)	Macro-F1 (%)
24 h	335	11	15	90.99	89.31	89.56	90.75	92.35	91.20
24 h-cleaned	254	7	8	92.32	91.52	90.57	92.70	94.76	93.30
72 h	323	7	10	91.71	90.55	90.70	93.72	93.65	93.64
72 h-cleaned	243	7	5	92.68	91.92	91.67	95.78	95.87	95.80
120 h	358	8	9	91.21	90.92	90.91	94.80	94.90	94.78
120 h-cleaned	270	6	5	91.80	90.97	91.00	96.34	96.30	96.31
168 h	351	6	7	92.21	92.55	91.45	96.12	95.99	96.03
168 h-cleaned	265	5	3	93.64	94.24	92.95	97.84	97.93	97.84

LVs, the number of latent variables; MDs, the number of misdiagnoses.

## Data Availability

The datasets presented in this article are not readily available because the data are part of an ongoing study. Requests to access the datasets should be directed to Z.H.

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
