# Peer review of "Selection of Optimal Diagnostic Positions for Early Nutrient Deficiency in Cucumber Leaves Based on Spatial Distribution of Raman Spectra"

_plants, 2025, doi:10.3390/plants14081199_

Round 1
Reviewer 1 Report
Comments and Suggestions for Authors
Title: Selection of Optimal Diagnostic Positions for Early Nutrient Deficiency in Cucumber Leaves Based on Spatial Distribution of Raman Spectra
# This manuscript addresses an important challenge in precision agriculture - the early diagnosis of nutrient deficiencies in cucumber leaves using Raman spectroscopy. This is highly relevant given the need for rapid, non-destructive methods in modern crop management. The experimental design is well described. The authors detail plant growth conditions, stress induction, spectral data acquisition, pre-processing and model evaluation (using PLS-DA and cosine similarity analysis). The combination of chemical analysis with spectral methods is clearly a strength. The use of dot matrix spectral sampling, box plots for outlier detection, heat maps for spatial distribution and comprehensive tables (e.g. Tables 1-4) are commendable. These contribute to a solid demonstration of how spectral variability can be managed to improve diagnostic accuracy. The paper clearly states the objective of selecting optimal diagnostic positions on cucumber leaves to improve early detection of N, P and K deficiencies. The conclusions drawn are supported by both visual and statistical evidence.
However, although the manuscript is detailed, there are several instances of awkward phrasing and minor grammatical issues that could be improved for clarity. In some sections, sentences are too long and could benefit from being broken up into shorter, more digestible chunks. In the introduction and methods sections, some transitions between ideas feel abrupt. Clearer bridges between concepts (e.g. when moving from chemical nutrient analysis to spectral data acquisition) would help guide the reader. While the methods are generally well described, some parameters (e.g. the rationale for selecting a 1.0 cm spacing template and its impact on sampling uniformity) could be more thoroughly justified. Although the figures are informative, some captions are overly detailed. Simplifying captions and ensuring that each figure and table is directly referenced and discussed in the text would improve readability. The discussion section could further explore the implications of the results for wider agricultural applications and mention limitations (such as potential environmental variability in field conditions) and future work.
# Mistakes and suggestions by lines
-Lines 16-17. “...for early detection of nitrogen (N), phosphorus (P), and potassium (K) deficiencies, providing a scientific basis for high-throughput phenotypic analysis via Raman spectroscopy (RS).”
Consider rephrasing for clarity. For example: “...for early detection of N, P, and K deficiencies, thereby providing a robust scientific basis for high-throughput phenotyping using Raman spectroscopy.”
-Lines 25–27. “...with lower similarity spectral data primarily concentrated near the leaf margins, main veins, and the leaf base.”
It might be clearer to state, “...with the lowest similarity values predominantly observed at the leaf margins, near the main veins, and at the leaf base.”
-Lines 29–30. “Excluding low-similarity data significantly improved model performance for early (24 h) nutrient deficiency diagnosis...”
Specify briefly what “improved model performance” means (e.g., “...resulting in higher precision, recall, and F1 scores”) to immediately highlight the benefits.
-Lines 38–40. The introduction effectively motivates the need for non-destructive methods. However, the sentence structure could be simplified. Break up complex sentences. For example, “Precision agriculture requires rapid, reliable, and non-destructive methods to capture crop information. Nutrient diagnostics play a pivotal role in this transition.”
-Lines 53–59. The description of traditional methods is clear, but consider highlighting their drawbacks more succinctly: “Although laboratory-based chemical analyses (e.g., Kjeldahl, spectrophotometry) offer high accuracy, they are time-consuming, destructive, and risk environmental contamination.”
-Lines 80–88. The rationale for focusing on optimal diagnostic positions is well explained. Rework the last sentence to improve clarity. For example, “Therefore, identifying specific positions on the leaf that yield the most stable and representative Raman spectra is critical for improving diagnostic accuracy in early nutrient deficiency detection.”
-Lines 107–117. The description of plant growth conditions is thorough. In line 116, consider clarifying why the 2-leaf, 1-core stage was chosen, linking it to the sensitivity of nutrient stress at this developmental stage.
- Lines 142–148. The Raman spectroscopy setup is described in detail. Mention briefly the justification for choosing a 785 nm laser (e.g., “which minimizes fluorescence interference”) to strengthen the methodology section.
-Lines 149–156. The selection of the dot-matrix template is a crucial detail. Explain the trade-offs (e.g., “Larger hole diameters facilitate data acquisition, but may reduce spatial resolution”) to help readers understand the decision process.
-Lines 184–211. The description of the diagnostic model development is comprehensive. In lines 203–210, clarify the criteria for choosing the optimal number of latent variables (LVs) and provide a brief explanation of why reducing misdiagnoses (MDs) is critical for model performance.
-Lines 216–224. The results on moisture and weight are well presented. It would be beneficial to include a brief interpretation of why moisture content follows the same trend as insect weight, linking it to the physiological responses of the insects.
-Lines 252–271. The explanation of cosine similarity is clear. In lines 265–270, discuss the practical implications of slight declines in similarity as the number of spectra increases. Explain how this informed the choice to focus on values below 0.991.
-Lines 317–377. The comparison of model metrics before and after data cleansing is detailed and informative. Consider summarizing key numerical improvements in a brief sentence to emphasize the impact of the data cleansing process on diagnostic accuracy.
- Lines 386–393. The discussion begins by framing the novelty of the approach in the context of existing literature. Strengthen this section by comparing your findings more explicitly with previous studies (e.g., “In contrast to Hu et al. [20], our study demonstrates…”).
-Lines 400–409. The authors discuss the spatial distribution of spectral similarity. Add a sentence that discusses potential limitations (e.g., “However, the variability in leaf size and environmental conditions may limit the generalizability of these spatial patterns”).
-Lines 421–431. Future directions are mentioned briefly. Expand on this by suggesting specific follow-up experiments (e.g., “Future work should consider multi-crop trials and real-field conditions to validate the robustness of these diagnostic positions”).
-Lines 432–447. The conclusions summarize the work well and restate the importance of excluding unstable spectral regions. Reiterate the broader impact on precision agriculture. For instance, “This method not only enhances early nutrient deficiency detection in cucumbers but also lays the groundwork for non-destructive diagnostics in other crops.”
# Global suggestions:
-Ensure that abbreviations like “RS” for Raman spectroscopy and “ND, PD, KD, CK” are consistently defined and used throughout the manuscript.
-There are several minor punctuation issues (e.g., missing commas in compound sentences) that should be addressed during final proofreading.
- Verify that all figures (e.g., Figures 1, 3, 4, 5, 6, 7) are referenced in the text and that their captions clearly explain the content without unnecessary repetition.
Comments on the Quality of English Language
Please, see report
Reviewer 2 Report
Comments and Suggestions for Authors
Dear all,
The manuscript, "Selection of Optimal Diagnostic Positions for Early Nutrient Deficiency in Cucumber Leaves Based on Spatial Distribution of Raman Spectra," explores a novel approach for diagnosing early nitrogen (N), phosphorus (P), and potassium (K) deficiencies in cucumber leaves using Raman spectroscopy (RS). The study employs a dot-matrix scanning method to analyze spectral similarity variations at different leaf positions and determines optimal diagnostic sites for early nutrient deficiency detection. The results suggest that regions near leaf margins, main veins, and the leaf base produce unstable spectral responses and should be avoided for accurate diagnosis. The findings enhance the precision of nutrient stress detection and offer valuable insights for precision agriculture and non-destructive plant health monitoring.
Strong Points
The manuscript presents an innovative approach to early nutrient deficiency diagnosis using RS, which is a non-invasive, real-time technique that could significantly advance precision agriculture. The study successfully integrates spectral similarity analysis and machine learning techniques to refine nutrient deficiency detection, improving the accuracy of plant health monitoring. The methodological approach is rigorous, involving multiple stress durations and detailed spatial distribution analysis, which strengthens the reliability of the findings. Furthermore, the research contributes to the broader application of RS in plant diagnostics and nutrient management, an area with growing significance in sustainable agriculture.
Weaker Aspects
In my opinion, the primary limitation of the study is the lack of a clearly stated rationale hypothesis. While the study identifies optimal diagnostic positions for RS-based nutrient stress detection, it does not explicitly define an initial hypothesis that guides the investigation. Additionally, the experimental design could be strengthened by providing details on the sources and concentrations of micronutrients in the nutrient solution, as these elements may influence spectral responses. The discussion should also elaborate on how spectral similarity variations correspond to physiological and biochemical changes under early nutrient stress. Lastly, the manuscript would benefit from a more detailed justification for the exclusion of certain spectral data points and how this affects model robustness.
So, I have annotated bellow and in the attached MS an attempt to clarify certain ideas, but the authors should examine my suggested wording changes carefully, to be sure that I have not misinterpreted what they wanted to say.
Abstract
-
Page 1, Line 16: Notice: The study should explicitly mention its contribution to precision agriculture.
Suggested Fix: "This study enhances the accuracy of Raman spectroscopy-based nutrient diagnosis, improving its application in precision agriculture." -
Page 1, Line 23: Notice: The broader agricultural implications of the findings should be emphasized.
Suggested Fix: "These results provide critical insights for developing rapid, non-destructive methods for nutrient stress monitoring in crops."
Introduction
-
Page 2, Line 43: Notice: The classification of N, P, and K as "primary macronutrients" may be misleading.
Suggested Fix: "N, P, and K are essential nutrients for plant growth, each playing critical roles in metabolism and development." -
Page 2, Line 70: Notice: The manuscript should address the potential of Raman spectroscopy to detect biochemical changes under hidden (latent) starvation.
Suggested Fix: "While Raman spectroscopy is effective in detecting visible nutrient deficiencies, its potential to diagnose latent starvation remains an important research area."
Materials and Methods
-
Page 4, Line 108: Notice: The rationale hypothesis is not clearly defined.
Suggested Fix: "We hypothesized that spatial spectral variations in leaves could be systematically analyzed to determine optimal diagnostic positions for early nutrient deficiency detection." -
Page 5, Line 122: Notice: The source and concentrations of micronutrients in the nutrient solution are missing.
Suggested Fix: "The nutrient solution included micronutrient supplements, consisting of [list specific elements and concentrations], to ensure comprehensive plant nutrition."
Results
-
Page 7, Line 222: Notice: The transition from a "sink" to a "source" is well described, but the nutrient concentration should be reported in SI-compliant units.
Suggested Fix: "Nutrient content should be reported in g kg⁻¹ instead of % to align with the International System of Units (SI)." -
Page 9, Line 310: Notice: The explanation of spectral similarity variations lacks a direct link to physiological responses.
Suggested Fix: "The observed spectral variations likely correspond to biochemical adjustments in leaf tissues during early nutrient stress adaptation."
Discussion
-
Page 13, Line 390: Notice: The exclusion of spectral data should be more thoroughly justified.
Suggested Fix: "Removing low-similarity data improved model accuracy by eliminating spectral noise and inconsistent signals from unstable leaf regions." -
Page 14, Line 425: Notice: The broader applicability of the diagnostic method under field conditions should be addressed.
Suggested Fix: "Future studies should validate the model across diverse environmental conditions to enhance its practical use in real-world agricultural settings."
Conclusion
- Page 15, Line 440: Notice: The conclusion should emphasize the potential integration of RS diagnostics with automated agricultural monitoring systems.
Suggested Fix: "Integrating this diagnostic approach with automated sensor networks could facilitate real-time, large-scale monitoring of plant nutritional status."

Reviewer 3 Report
Comments and Suggestions for Authors
Summary:
The author selected a particular cucumber variety as a case model and designed variable control study to induce nutrition deficiency in the plant and demonstrate the possibility of using non-destructive Raman Spectrometry in plant nutritional monitoring. To further improve the throughput and reliability of the method, the author identified key regions on the cucumber leaves that can provide spectrum of higher similarity to the mean spectrum, and built prediction models to different stress situations.
This is a preliminary study in controlled lab environment using destructive(non-alive) samples from a particular plant species, the practicality of the method on more plant species seems in real field application seems still far and needs more validation. I do have the following comments that need to be addressed.
- In line 124-126, the nutrient deficiency solution design was discussed. For the PD nutrient solution, is there a substitute for the replaced (NH₄)H₂PO₄? And could you specify the concentration for the substituted compounds? For example, in the ND nutrient solution, at what concentration are CaCl₂ and KCl added?
- In line 129-131, the author described the first-node leaves were excised for RS collection. How long is the sample transport time (hours?days?) in between between leaves excision and arriving at the lab for RS and chemical analysis?
- In line 63, the author suggests Raman spectroscopy is a non-destructive technique. And in line 143, the author confirmed a miniature Raman spectrometer was used. Why the experimental design still employed destructive leaves excision approach to collect spectrum if the purpose is to demonstrate the advantages of RS as a non-destructive technique?
- In line135-136:”…placed in an oven at 105°C for inactivation, followed by drying at 80°C to a constant weight.” This needs more explanation. Is the sample placed in oven at 105°C for enzymatic inactivation or other inactivation? Why has to be 105°C? Wouldn’t 80°C be suffice for halting enzymatic inactivation?
- In line 137-140. The author briefly mentioned the method for each nutrient content measurement. More quantitative description or appropriate reference to the method would be necessary for interested readers to reproduce the experiment.
- In line 231-233, the author described the N, P, K content in CK group to leaves to exhibit an initial increase followed by a decline. From Figure 2, only nitrogen content clearly displayed such behavior, whereas such trend is hard to claim for P and K given the large standard deviation and small sample set (n=3). A more accurate discussion on the Figure 2 CK group data here is preferred.
- In line 268-269, the author described a spectrum similarity score threshold of 0.991 determined in the lab environment and spectrometer condition specific to the author at the time of the experiment. As discussed by the author in line 266-267, the similarity score trend may vary due to environmental interferences or the stability of the spectrometer. Then how could this one-time determined threshold value be used for subsequent experiments? Additionally, what is the actual time difference in between acquiring 10, 20, 30 spectra? Based on line 146-148, each spectrum is the mean of three consecutive acquisitions, with each acquisition has an integration time of 3.5 seconds. Then every 10 spectra would take 105 seconds. Then this cumulative effect of environmental interference in controlled lab environment or spectrometer instability on spectral quality similarity decline is evident in less than every 2 minutes, how many spectra could then be collected in real field applications where portal spectrometers are taken to the field for non-destructive sampling on leaves with no sample fixation or preparation?
Round 2
Reviewer 1 Report
Comments and Suggestions for Authors
The manuscript has been significantly improved, so it could be published in its current form.
Reviewer 2 Report
Comments and Suggestions for Authors
Dear All,
After thoroughly analyzing the revised version of the manuscript and the authors' detailed rebuttal letter, I affirm that the authors have carefully addressed all previously raised concerns. The revised manuscript demonstrates significant improvement in clarity, technical detail, and scientific rigor.
The inclusion of clear hypotheses, the explanation of the Hoagland solution micronutrient composition, and the corrections to SI unit conventions (e.g., g·kg⁻¹ instead of %) greatly enhanced transparency and reproducibility. Furthermore, the integration of new insights—such as the spatial instability of Raman signals near leaf margins, veins, and bases—is both scientifically valuable and practically relevant for precision agriculture.
The manuscript now more explicitly positions its contribution to precision agriculture, particularly in the context of non-destructive diagnostics. The improved conclusion now also highlights potential integration with sensor networks, which broadens the real-world applicability of the findings.
I appreciate the authors’ thoughtful revisions and commend their responsiveness. Therefore, I am happy to endorse this manuscript for publication.
Kind regards,
Reviewer 3 Report
Comments and Suggestions for Authors
Thanks to the author for have successfully addressed my comments in great detail.